# $E(3) \times SO(3)$-Equivariant Networks for Spherical Deconvolution in Diffusion MRI

**Axel Elaldi**[1]                               AXEL.ELALDI@NYU.EDU
**Guido Gerig**[1]                                 GERIG@NYU.EDU
**Neel Dey**[2]                                   DEY@MIT.EDU
[1] *VIDA Center, Computer Science and Engineering, New York University*
[2] *Computer Science and Artificial Intelligence Lab, Massachusetts Institute of Technology*

**Editors:** Accepted for publication at MIDL 2023

## Abstract

We present Roto-Translation Equivariant Spherical Deconvolution (RT-ESD), an $E(3) \times SO(3)$ equivariant framework for sparse deconvolution of volumes where each voxel contains a spherical signal. Such 6D data naturally arises in diffusion MRI (dMRI), a medical imaging modality widely used to measure microstructure and structural connectivity. As each dMRI voxel is typically a mixture of various overlapping structures, there is a need for blind deconvolution to recover crossing anatomical structures such as white matter tracts. Existing dMRI work takes either an iterative or deep learning approach to sparse spherical deconvolution, yet it typically does not account for relationships between neighboring measurements. This work constructs equivariant deep learning layers which respect to symmetries of spatial rotations, reflections, and translations, alongside the symmetries of voxelwise spherical rotations. As a result, RT-ESD improves on previous work across several tasks including fiber recovery on the DiSCo dataset, deconvolution-derived partial volume estimation on real-world *in vivo* human brain dMRI, and improved downstream reconstruction of fiber tractograms on the Tractometer dataset. Our implementation is available at https://github.com/AxelElaldi/e3so3_conv.

**Keywords:** Equivariant Networks, Diffusion MRI, Spherical Deep Learning

## 1. Introduction

Diffusion MRI (dMRI) is widely used for imaging water diffusion within the brain by measuring diffusion rate over the unit sphere at each voxel. Specialized dMRI algorithms operating on voxel-wise spheres can recover the neuronal tracts and structural organization of the brain. However, as each voxel may contain overlapping microstructures (e.g., crossing tracts) and is subject to both spatial and spherical partial voluming, a blind source separation problem arises at each voxel. This paper identifies two key limitations of existing dMRI deconvolution work and presents an unsupervised geometric deep learning approach to recover unmixed per-voxel *fiber orientation distribution functions* (fODFs) from dMRI.

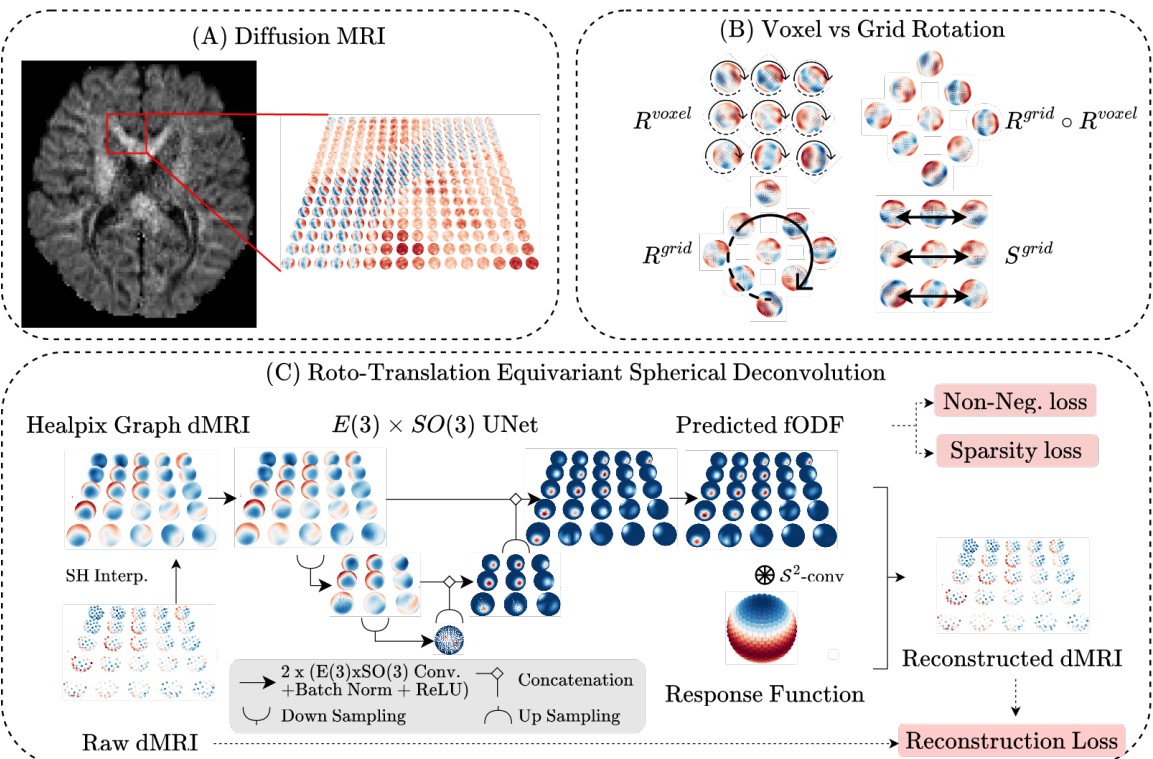

Figure 1: **Motivation and methods overview. A.** Diffusion MRI measures a spatial grid of spherical signals. **B.** In addition to translations and grid reflections, we construct layers equivariant to voxel and grid-wise rotations and any combination thereof. **C.** RT-ESD takes a patch of spheres and processes it with an $E(3) \times SO(3)$-equivariant UNet to produce fODFs. It is trained under an unsupervised regularized reconstruction objective.

**Spherical deconvolution: linear and nonlinear.** Voxelwise spherical signals are often assumed to be a convolution between an fODF (a non-negative spherical function indicating neuronal fiber direction and intensity) and a tissue response function (a spherical point spread function), which in turn motivates the inverse recovery of the fODF. Several regularized iterative optimization-based methods have been developed to this end, yet they typically estimate fODFs linearly and can struggle to resolve fibers crossing at small angles. Recent progress has been made by using deep networks to regress fODFs in a supervised manner using either ex-vivo histology data (Nath et al., 2019) or assuming previous iterative model fits (Jeurissen et al., 2014) to be 'ground truth' targets (Nath et al., 2020; Sedlar et al., 2020). However, such approaches are limited in that they require ex vivo training data and are upper bounded by the performance of Jeurissen et al. (2014). More recently, an unsupervised rotation-equivariant spherical deconvolution network (ESD) was proposed in (Elaldi et al., 2021), yet their approach only performs voxel-wise independent operations.

**Spatial coherence.** Neighboring voxels are likely to yield similar fODF estimates. Nevertheless, most dMRI methods deconvolve fODFs in an independent voxel-wise manner with few exceptions. These include spatial regularization via total variation (Canales-Rodríguez

et al., 2015) and fiber continuity/regularity (Goh et al., 2009; Ramirez-Manzanares et al., 2007; Reisert and Kiselev, 2011). However, current deep networks do not explicitly model inter-voxel dependence (beyond preliminary attempts with channel-wise concatenation of a voxel neighborhood) and we speculate that deep unsupervised fODF estimation can be further improved with spatial information and *spatio-spherical* weight-sharing.

**Mis-specified inductive biases.** Standard convolutional networks for scalar images are equivariant to the translation group (up to aliasing) and this weight-sharing is crucial to their strong generalization. For data living on the sphere, *rotation*-equivariant convolutions can be defined analogously. However, for dMRI, there are currently no deep networks that *simultaneously* respect the symmetries of pointwise rotations and spatial rotations, translations, and reflections (see Fig. 1) which would enforce the network output to change predictably under these transformations, thus increasing robustness and performance.

**Contributions.** To nonlinearly process dMRI with the correctly specified spatio-spherical inductive biases, this work develops convolutional networks for inputs living in $\mathbb{R}^3 \times \mathcal{S}^2$ with layers that are equivariant to the $E(3) \times SO(3)$ group. Consequently, these layers are arranged in an image-to-image architecture (RT-ESD) to recover sparse, unmixed, and spatially-coherent fODFs in an unsupervised manner. Quantitatively, these developments lead to improved recovery of fibers in various synthetic and challenge datasets and improved partial volume estimation in in-vivo human data without known ground truth.

## 2. Related work

**fODF Deconvolution.** Constrained Spherical Deconvolution (Jeurissen et al., 2014; Tournier et al., 2007) (CSD) is the most widely-used fODF recovery method for its simplicity and performance on dMRI with dense angular sampling. Further, several extensions of CSD using dictionaries (Filipiak et al., 2022) and various forms of regularization (e.g., sparsity (Canales-Rodríguez et al., 2019) and spatial coherence (Canales-Rodríguez et al., 2015; Goh et al., 2009; Ramirez-Manzanares et al., 2007; Reisert and Kiselev, 2011)) have been developed for more specific use-cases. More recently, deep regression networks have been trained to reproduce CSD fits (Jha et al., 2022; Nath et al., 2020) for improved speed and tolerance to undersampling with similar work concatenating neighborhood voxels along input-channels (Lin et al., 2019; Sedlar et al., 2020) with no spatial weight-sharing.

**Equivariant deep learning.** Relevant to our spatial desiderata, $SE(3)$ or $E(3)$ equivariant networks have been developed for volumes (Weiler et al., 2018), point clouds (Thomas et al., 2018), meshes (Suk et al., 2022), and graphs (Zhao et al., 2021; Brandstetter et al., 2022), among others. With respect to spherical data, $SO(3)$-equivariance can be obtained with harmonic methods (Cohen et al., 2018; Kondor et al., 2018; Ocampo et al., 2023) or via isotropic Laplacian convolutions on spherical graphs (Perraudin et al., 2019), the latter of which we extend to $\mathbb{R}^3 \times \mathcal{S}^2$ dMRI inputs for $E(3) \times SO(3)$ equivariance.

**Equivariant learning for dMRI.** Most relevant to this work, ESD (Elaldi et al., 2021) trains an unsupervised per-voxel $SO(3)$-equivariant UNet to recover fODFs in a voxelwise manner and RT-ESD can be thought of as a spatial generalization of ESD. Outside of deconvolution, Banerjee et al. (2020); Bouza et al. (2021) develop convolutions that exhibit

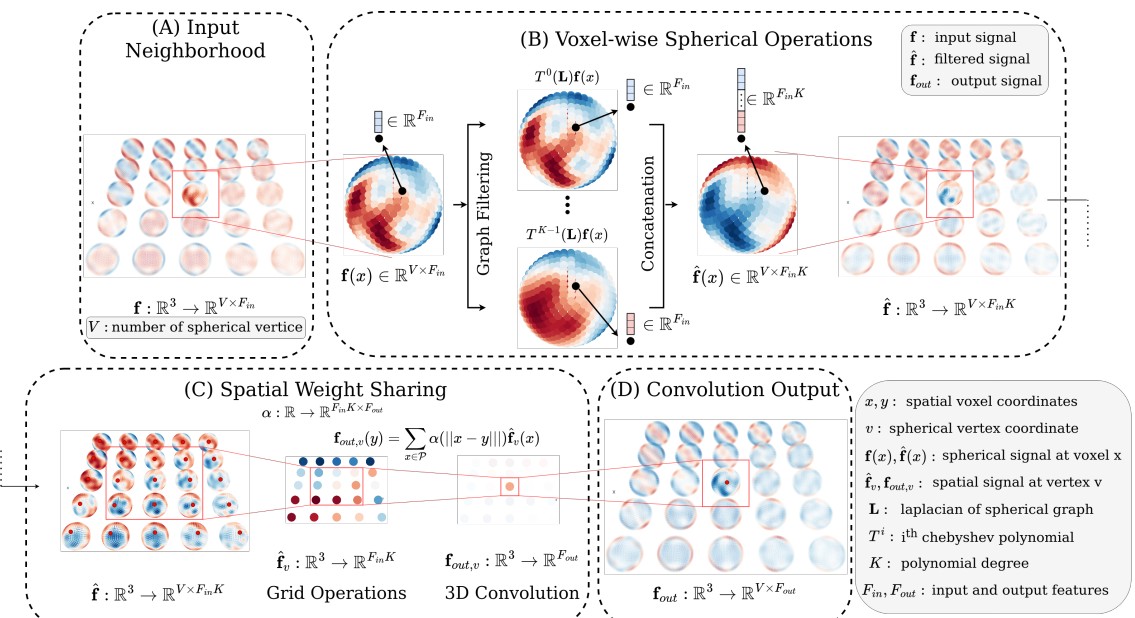

Figure 2: **E(3) × SO(3) Convolutions**. (a) The input is a patch of spherical signals $\mathbf{f}$ with $F_{in}$ features. For each voxel $x \in \mathbb{R}^3$, $\mathbf{f}(x)$ is projected onto a spherical graph $\mathcal{G}$ with $V$ nodes. (b) The convolution first filters each sphere with Chebyshev polynomials applied to the Laplacian $L$. The filter outputs are then aggregated along the grid to create a spherical signal $\hat{\mathbf{f}}$ with $F_{in}V$ features. (c) For each $v \in \mathcal{G}$, we extract the corresponding spatial signal $\hat{\mathbf{f}}_v(\cdot)$. (d) These $V$ convolutions give the final grid of spheres $\mathbf{f}_{out}$. Connected boxes across (c) and (d) show spatial operations on a single spherical vertex.

voxel-wise $SO(3)$-equivariance and incorporate manifold-valued spatial averaging that yields improved dMRI classification and super-resolution. $SO(3)$-networks have also seen success in regressing dMRI scalar maps (Goodwin-Allcock et al., 2022; Sedlar et al., 2021), and tractography (Sinzinger and Moreno, 2022). Lastly, SE(3)-equivariance for improved $\mathbb{R}^3 \times \mathcal{S}^2$ dMRI segmentation has been achieved via memory-intensive tensor field networks (Müller et al., 2021) and separable kernels (Liu et al., 2022). In contrast, we focus on fODFs and equivariance to both joint and *independent* voxel and grid rotations.

## 3. Methods

**Background.** We follow dMRI deconvolution, multi-tissue, multi-shell, and response function conventions from Elaldi et al. (2021). Given $f : \mathbb{R}^3 \times \mathcal{S}^2 \to \mathbb{R}$, we denote $f(x,.) = \mathbf{f}(x) : \mathcal{S}^2 \to \mathbb{R}$ where $x$ is the spatial coordinate. Grid rotations by angle $p$ are $R_p^{\text{grid}} f(x,q) = f(R_p^{-1}x, q)$ and voxelwise rotations are $R_p^{\text{voxel}} f(x,q) = f(x, R_p^{-1}q)$ where $q$ is the spherical coordinate. A network $\mathcal{N}$ is grid and/or voxelwise-rotation equivariant if $R_p^{\text{grid}} \mathcal{N}(f) = \mathcal{N}(R_p^{\text{grid}} f)$ and/or $R_p^{\text{voxel}} \mathcal{N}(f) = \mathcal{N}(R_p^{\text{voxel}} f)$, respectively.

The dMRI signal $S$ is equal to the spherical convolution of the fODF $F$ with tissue-specific response function $R : \mathcal{S}^2 \to \mathbb{R}^B$ where $B$ is the number of shells. $F$ is voxel-dependent and rotationally symmetric, while $R$ are shell-dependent and symmetric about

the y-axis. W.r.t. spatio-spherical convolutions, let $\psi : \mathbb{R}^3 \times \mathcal{S}^2 \to \mathbb{R}$ be a spherical filter, such that convolving $f$ and $\psi$ yields $f_{out}(x, q) = \int_{\mathbb{R}^3} \int_{\mathcal{S}^2} \psi(x - y, R_p^{-1} q) f(y, p) dp dy$ where $x, y \in \mathbb{R}^3$ and $p, q \in \mathcal{S}^2$ are voxel and spherical coordinates, respectively, and $R_p$ is the rotation matrix. We reduce this to a sequential spherical and spatial convolution below.

**Spherical convolutions.** Spherical deep learning applies convolutions between a spherical signal and learnable spherical filters, with rotation-valued output features living on **SO**(3) (Cohen et al., 2018). For speed and memory efficiency, we approximate spherical convolutions using graph convolutions with isotropic kernels (Perraudin et al., 2019). We discretize $f(x, .)$ on vertices $\mathcal{V} = (p_i)_i \in (\mathcal{S}^2)^{|\mathcal{V}|}$. For all $x \in \mathbb{R}^3$, we have $\mathbf{f}(x) = (f(x, p_i))_i \in \mathbb{R}^{|\mathcal{V}|}$. From $\mathcal{V}$, we construct a graph $\mathcal{G} = (\mathcal{V}, w)$, where $w$ are the edge weights. We use the graph convolution $\mathbf{f}_{out}(x) = h(\mathbf{L})\mathbf{f}(x) = (\sum_{k=0}^{K-1} \alpha_k \mathbf{L}^k)\mathbf{f}(x)$ where $\mathbf{L}$ is the Laplacian, $(\alpha_k)_k \in \mathbb{R}^K$ are learnable weights, and $K$ is the polynomial degree of the convolution. Practically, we compute the $K$ laplacian polynomials $T^k(\mathbf{L})\mathbf{f}(x)$ using Chebyshev polynomials.

**E(3) $\times$ SO(3) convolutions.** Let the spatial component of $f$ be sampled on point cloud $\mathcal{P} = (q_i)_i \in (\mathbb{R}^3)^{|\mathcal{P}|}$. As *anisotropic* SE(3)-equivariant point cloud convolutions have intractable time and memory complexity for large dMRI volumes, we use isotropic $SE(3)$-point cloud filters by using Thomas et al. (2018) with only a scalar field. That is, for roto-translation equivariant convolutions, the filter $\alpha_k$ depends only on the norm of the points, $\alpha_k(x) = \alpha_k(||x||), \forall x \in \mathbb{R}^3$, such that, $\mathbf{f}_{out}(x) = \sum_{y \in \mathcal{P}} \sum_{k=0}^{K-1} \alpha_k(||x - y||)\hat{\mathbf{f}}_k(y)$ , where $\alpha_k : \mathbb{R}^+ \to \mathbb{R}$ is a learnable isotropic kernel, and $\hat{\mathbf{f}}_k(y) = T^k(\mathbf{L})\mathbf{f}(y)$ is the spherical graph filtering described above. For multichannel $f$, we have $\hat{\mathbf{f}}_k^c = T^k(\mathbf{L})\mathbf{f}^c$ and $\mathbf{f}_{out}^{c'}(x) = \sum_m \sum_{y \in \mathcal{P}} \alpha^{m,c'}(||x - y||)\hat{\mathbf{f}}^m(y)$ where $c$ is the channel index and $m = (c, k)$. Practically, this is implemented using a 3D convolution, with the weights shared across the $|\mathcal{V}|$ 3D filtered maps $\hat{\mathbf{f}}_{k,v}$ where $v \in \mathcal{V}$ are vertices.

**E(3) $\times$ SO(3)-equivariance.** The proposed convolution is $E(3)$-equivariant to the grid transformation $\tau \in E(3)$ as $\tau$ preserves the distance between two points $x, y \in \mathbb{R}^3$ and $\alpha^{m,c'}$ is isotropic, i.e. , $\alpha^{m,c'}(||\tau^{-1}x - y||) = \alpha^{m,c'}(||x - \tau y||)$. Further, a voxel rotation $R^{voxel}$ on $\mathbf{f}_{out}^{c'}$ acts only on the spherical outputs $\hat{\mathbf{f}}_k$. Thus, voxel-wise $SO(3)$-equivariance corresponds to the $SO(3)$-equivariance of $T^k(\mathbf{L})\mathbf{f}$, originally developed in (Perraudin et al., 2019).

**Roto-Translation Equivariant Sparse Deconvolution.** RT-ESD (Fig 1c) deconvolves dMRI using a UNet with $E(3) \times SO(3)$-equivariant layers. The inputs $f(x, .)$ are typically sampled with a few dozen to a few hundred protocol-dependent directions which are interpolated onto a HEALPix grid following Elaldi et al. (2021). Architecturally, we use the same UNet as Elaldi et al. (2021) and replace its layers with ours. Further, as up/downsampling layers have to be adapted to $\mathbb{R}^3 \times \mathcal{S}^2$, we decompose (un)pooling into a mean spatial (un)pooling on $\mathbb{R}^3$ followed by a mean spherical (un)pooling on $\mathcal{S}^2$. To minimize the equivariance error from the spherical pooling, we use the hierarchical structure of the HEALPix grid. Batch normalization and pointwise ReLUs are used after every convolution. The network estimates one fODF per tissue compartment.

The estimated fODFs are subsequently convolved with tissue response functions estimated with Tournier et al. (2019) to yield the reconstructed dMRI signal. We train the UNet $\mathcal{N}$ to recover fODF $F(x) = \mathcal{N}(S(x))$ by minimizing the unsupervised regularized

reconstruction loss:

$$\mathcal{L} = \|S(x) - (R * \mathcal{N}(S(x)))\|_2^2 + \lambda Reg(\mathcal{N}(S(x)))$$

where $Reg$ is the sparse and non-negative regularizer on $F$ from Elaldi et al. (2021).

## 4. Experiments

Due to unknown fiber distributions in human dMRI, we focus our quantitative evaluations on synthetic and benchmark data with underlying ground truth fODFs and/or tractography. W.r.t. *in vivo* human dMRI, we evaluate unsupervised tissue partial volume estimation as a surrogate for deconvolution performance as in Elaldi et al. (2021). Our deconvolution baselines include the optimization-based CSD (Tournier et al., 2019), a voxelwise deconvolution network (ESD) (Elaldi et al., 2021), and a spatial extension of ESD inspired by Sedlar et al. (2020) where neighboring voxels in a patch are concatenated channelwise (C-ESD). The numerical equivariance of the proposed layers is benchmarked against previous work in App. A.1, downstream tractography evaluations on Tractometer (Maier-Hein et al., 2017) are presented in App. A.3, and additional implementation details are provided in App. B.

### 4.1. Simulated data: $\mathbb{R}^3 \times \mathcal{S}^2$ MNIST segmentation

**Data.** To evaluate the generic utility of $E(3) \times SO(3)$-equivariant layers, analogous to spherical MNIST (Cohen et al., 2018), we simulate an $\mathbb{R}^3 \times \mathcal{S}^2$ version of MNIST where images are projected to a $16 \times 16 \times 16$ grid of spheres where each sphere contains a projected MNIST image and the spheres are spatially correlated in their classification labels. This is done to isolate and study the spatio-spherical components of network layers in a voxel-classification/spatial segmentation setting. Sample data and labels are illustrated in Figure 3A and B and the simulation design choices are detailed in Appendix B.2. As voxelwise MNIST classification is trivially achieved, the classification labels for the spheres are constructed to be spatially correlated. We project random image *crops* to the sphere, such that spatial dependencies need to be learned for high classification performance. To study generalization encouraged by equivariance, we then augment this data into four different datasets with grid rotations alone, with voxelwise rotations alone, with independent grid and voxel rotations, and the original untransformed images. 716/142/142 train/validation/test images are simulated for each dataset.

**Evaluation.** As E(3)-equivariant baselines, we use 3D CNNs with isotropic kernels trained on data with raw directional volumes or spherical harmonics coefficients concatenated channelwise (inspired by Nath et al. (2020)). As a voxelwise $SO(3)$-equivariant baseline, we use Perraudin et al. (2019). All methods are trained either on the untransformed data or on the data augmented with independent rotations and tested on each dataset separately to understand the gap between explicit equivariance and rotation augmentation.

**Results.** Trained without augmentation (Fig. 3C, left), all methods that incorporate *spatial dependency* perform well when evaluated on the untransformed data. However, unseen grid and/or voxel transformations severely reduce segmentation performance for all baselines with the developed $E(3) \times SO(3)$ network demonstrating high generalization to unseen

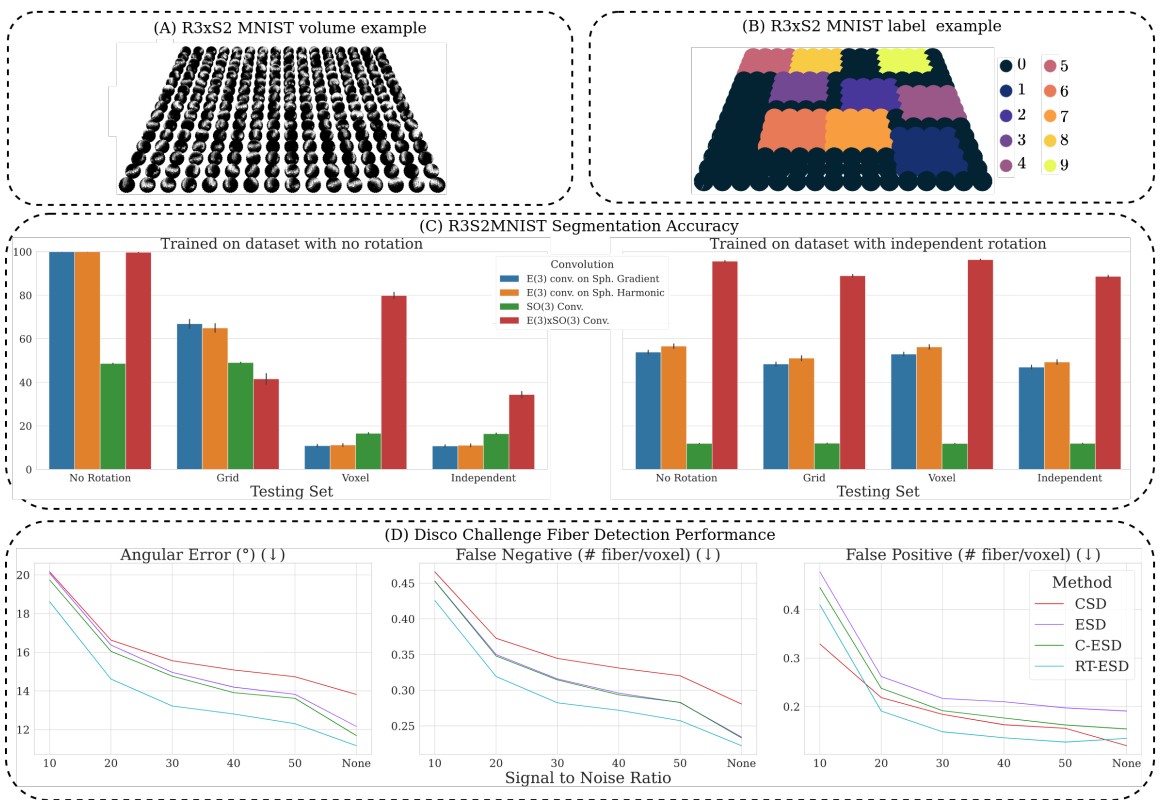

Figure 3: **A.** and **B.** visualize the spatio-spherical images and label maps for $\mathbb{R}^3 \times \mathcal{S}^2$ MNIST, respectively. **C.** Classification performances when trained on data with (**right**) or without (**left**) rotation augmentation and tested on data with no rotations, grid-rotations, voxel-rotations, and independent grid and voxel-rotations. **D.** Angular error and false positive/negative results on the DiSCo dataset (Sec 4.3) vs input SNR.

poses. When trained on a dataset with independent grid and voxel rotations (Fig. 3C, right), the developed method displays high performance and generalization w.r.t. baselines.

### 4.2. Benchmark data: DiSCo deconvolution and connectivity

**Data.** We assess local fODF reconstruction performance and robustness to diffusion MRI noise on the Diffusion-Simulated Connectivity (DiSCo) challenge dataset (Rafael-Patino et al., 2021). The DiSCo dataset has three $40 \times 40 \times 40$ volumes, each with six different noise levels ($SNR = 10, 20, 30, 40, 50, \infty$ dB). All volumes share the same protocol, with 4 B0 images and shells, and 60 gradients per shell. For each volume and noise level, we have access to the *ground truth fODF* and assume a three-tissue compartmentalization.

**Evaluation.** Deep learning results are averaged across 5 random seeds. fODFs are estimated by all baselines and fiber directions are estimated using DiPy (Garyfallidis et al., 2014). Our scoring follows Daducci et al. (2013) and is averaged across the volumes and presented for each SNR level. Detected fibers are matched to ground truth fibers using a rejec-

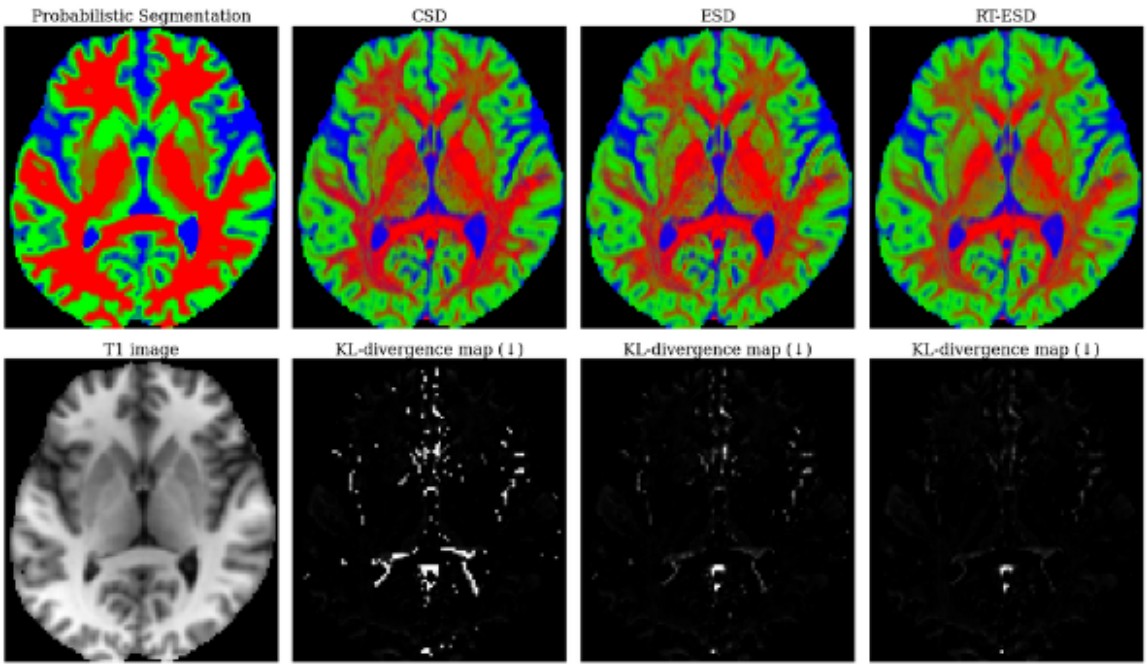

Figure 4: **Unsupervised partial volume estimation. Col. 1:** T1w MRI and label map of the subject co-registered to the dMRI input. **Cols. 2–4, row 1:** Partial volume estimates from each deconvolution method. **Cols. 2–4, row 2:** Divergence maps between the estimated partial volumes and the reference segmentation.

Table 1: KL-divergence (lower is better) on Partial Volume Estimation from the human dMRI dataset on three subjects (Sec. 4.3), averaged over 4 random seeds each.

| | CSD | ESD | Concat-ESD | | | RT-ESD | | |
|---|---|---|---|---|---|---|---|---|
| Patch size | $1^3$ | $1^3$ | $3^3$ | $5^3$ | $7^3$ | $3^3$ | $5^3$ | $7^3$ |
| Subj. 1 ($\downarrow$) | 1.28 | 0.62 | 0.62 | 0.94 | 0.61 | 0.65 | 0.61 | **0.56** |
| Subj. 2 ($\downarrow$) | 1.17 | 0.64 | 0.64 | 0.61 | 0.64 | 0.67 | 0.64 | **0.55** |
| Subj. 3 ($\downarrow$) | 1.53 | 0.82 | 0.99 | 0.88 | 0.86 | 0.80 | 0.85 | **0.72** |

tion cone of 25° and are used to compute false positive/negatives and angular errors. Lastly, we also compute the success rate (percentage of voxels with no false positives/negatives).

**Results.** Fig. 3 presents results using optimal spatial patch sizes for all methods, with patch size sensitivity studied in App. A.2. ESD performs better than CSD for SNR > 20 dB. At SNR ≤ 20 dB, CSD and ESD are similar. C-ESD with concatenated neighbors improves performance slightly. Lastly, RT-ESD improves angular error and false negative rates over all SNRs evaluated and also the false positives for all noise levels except for SNR=10 dB. At SNR=10, CSD outperforms every deep learning method in terms of success rate.

### 4.3. Human brain diffusion MRI partial volume estimation

**Dataset.** For direct comparison with Elaldi et al. (2021), we use the preprocessed multishell dataset from center 1 of Tong et al. (2020) consisting of three subjects each with 3 shells, 98 gradients per shell, and 27 B0 images. We use the grey/white matter and cerebrospinal fluid compartmentalization of the human brain for a three tissue decomposition.

**Evaluation.** As ground-truth fODFs, tractograms, and connectivities are unknown for *in vivo* data, our evaluation relies on a surrogate downstream task of unsupervised partial volume estimation (PVE) following Elaldi et al. (2021). For each tissue, we use the 0-degree spherical harmonic coefficient as the partial volume of that tissue compartment. We then compare the dMRI estimated PVE against a probabilistic 3-tissue segmentation of the co-registered high-quality T1w MRI using Zhang et al. (2001). The closer the PVE produced by each baseline matches the probabilistic segmentation, the better it deconvolves dMRI.

**Results.** KL-divergences between the reference and method-estimated PVE are given in Table 1 alongside visualizations in Fig. 4. The deep learning-based ESD and C-ESD methods improve on CSD, but still struggle at tissue interfaces. With increasing spatial patch size, we find that RT-ESD outperforms all previous baselines in tissue-specific deconvolution quality when using a spatial grid of $7 \times 7 \times 7$ voxels by leveraging spatial structure.

## 5. Discussion

**Limitations and future work.** Our experiments perform *instance-specific* optimization, which is time-consuming and suboptimal. Future work should consider training on large diffusion MRI datasets for amortized inference on unseen data. Further, due to a lack of ground truth, our quantitative *in vivo* evaluation is limited to evaluating surrogate tasks. We will therefore incorporate expert evaluations of tractograms in future work. Lastly, these layers can be easily integrated into other problems such as denoising and segmentation.

**Summary.** This paper developed convolutional layers that respect the structure of $\mathbb{R}^3 \times \mathcal{S}^2$ data and demonstrated their utility in diffusion MRI deconvolution. The proposed convolutions show improved robustness to unseen input transformations with increased spatial coherence leading to better anatomical recovery in terms of fiber scores and partial volume estimation over previous spherical deconvolution methods. These benefits were shown to be consistent across a segmentation task on simulated $\mathbb{R}^3 \times \mathcal{S}^2$ data and spatio-spherical deconvolution tasks on two challenge datasets and *in vivo* human brains.

### Acknowledgments

Axel Elaldi and Guido Gerig are supported by NIH grants 1R01HD088125-01A1, R01HD055741, 1R01MH118362-01, 2R01EB021391-06 A1, R01ES032294, R01MH122447, and 1R34DA050287. Neel Dey is supported by NIH NIBIB NAC P41EB015902 and NIH NIBIB 5R01EB032708.

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

## Appendix A. Additional results

### A.1. Numerical equivariance error

**Motivation.** As is common in discrete implementations of deep networks (Gruver et al., 2022), the proposed framework is not exactly equivariant. The spherical harmonic interpolation and convolution are voxel-wise rotation equivariant up to consistent spherical sampling and signal bandwidth and the up/downsampling and convolutional layers demonstrate inexact $E(3) \times SO(3)$ equivariance due to aliasing. We therefore quantify numerical rotation equivariance error for both one single convolutional layer and for the entire UNet with different convolution types.

**Evaluation.** We evaluate the equivariance of the developed method and baselines to grid rotations, voxelwise rotations, and simultaneous independent grid and voxelwise rotation equivariance. The considered methods include two types of convolutions on $\mathbb{R}^3$ data, one on $\mathcal{S}^2$ data, and the proposed convolution on $\mathbb{R}^3 \times \mathcal{S}^2$. They are the baselines used in Section 4.2. All convolutions use isotropic filters and are randomly initialized and frozen.

The `Gradient` convolutions apply 3D isotropic kernel convolutions on 3D directional volumes concatenated channelwise. Analogously, the `SH` convolutions apply 3D isotropic kernel convolutions on spatial grids of spherical harmonic fit coefficients concatenated channelwise. The `Spherical` convolutions treat each voxel independently with $SO(3)$-equivariant layers. We were unable to experiment with SE(3)-equivariant layers developed by Müller et al. (2021) due to intractable time complexity and GPU memory constraints. We use the equivariance error defined below and average it over a thousand random rotations:

$$\text{Equivariance error} = \frac{||Conv(R_p^{\text{grid}}(R_p^{\text{voxel}}(f))) - R_p^{\text{grid}}(R_p^{\text{voxel}}(Conv(f)))||_2^2}{||Conv(f)||_2^2}$$

**Results.** Results are provided in Table 2. They confirm that the four tested convolutions are grid rotation-equivariant due to the isotropic spatial filters. Further, the $SO(3)$ and $E(3) \times SO(3)$ convolutions are also approximately voxel-wise rotation equivariant. As a consequence, the full UNet architecture becomes more equivariant to grid and voxel rotation. It is important to note that equivariance error is compositional (Gruver et al., 2022) and therefore the equivariance error accumulated across the entire UNet is non-negligible. However, these trends are consistent with previous work in equivariant deep learning (Cohen et al., 2018).

Table 2: Mean rotation equivariance errors (with 95% CI) over different rotation types, averaged over 1000 random trials using an $8 \times 8 \times 8$ voxel grid with 192 points/sphere using a Healpix grid of **resolution 4** (192 points).

| | | Single convolutional layer | | | UNet | | |
|---|---|---|---|---|---|---|---|
| Group | Model | Grid Rot. | Voxel Rot. | Indep. Rot. | Grid Rot. | Voxel Rot. | Indep. Rot. |
| $E(3)$ | `Gradient` | $\mathbf{0.09 \pm 0.00}$ | $1.41 \pm 0.01$ | $0.92 \pm 0.01$ | $\mathbf{0.60 \pm 0.02}$ | $1.41 \pm 0.01$ | $1.26 \pm 0.01$ |
| $E(3)$ | `SH` | $\mathbf{0.09 \pm 0.00}$ | $1.98 \pm 0.01$ | $1.36 \pm 0.01$ | $\mathbf{0.60 \pm 0.02}$ | $1.43 \pm 0.01$ | $1.26 \pm 0.01$ |
| $SO(3)$ | `Spherical` | $\mathbf{< 0.01}$ | $\mathbf{< 0.01}$ | $\mathbf{< 0.01}$ | $0.89 \pm 0.04$ | $\mathbf{0.56 \pm 0.02}$ | $0.87 \pm 0.04$ |
| $E(3) \times SO(3)$ | `Ours` | $0.14 \pm 0.01$ | $\mathbf{< 0.01}$ | $\mathbf{0.14 \pm 0.01}$ | $0.71 \pm 0.03$ | $\mathbf{0.56 \pm 0.03}$ | $\mathbf{0.82 \pm 0.04}$ |

## A.2. DiSCo patch size sensitivity

Figure 5: An extended version of Figure 3 illustrating Angular Error, Success Rate, and False Negative and Positive Rates of predicted fODFs as a function of input image SNR for all baselines across all patch sizes on the DiSCo dataset (Sec. 4.2).

## A.3. Downstream tractography utility: Tractometer results

Table 3: Quantitative benchmarks on the Tractometer dataset derived from tractography downstream to fODF estimation. RT-ESD with a patch size of $3 \times 3 \times 3$ outperforms several baselines in terms of both tractography performance and partial volume estimation.

| Scores | CSD $1 \times 1 \times 1$ | ESD $1 \times 1 \times 1$ | C-ESD $3 \times 3 \times 3$ | RT-ESD $3 \times 3 \times 3$ |
|---|---|---|---|---|
| Valid Bundles ($\uparrow$) | 19 | 19 | 19 | **20** |
| Valid Streamline ($\uparrow$) | 72.2 | 72.3 | 73.0 | **77.3** |
| Mean Overreach ($\downarrow$) | **16.9** | 24.8 | 24.4 | 26.1 |
| Mean Overlap ($\uparrow$) | 59.8 | 65.4 | 65.4 | **65.7** |
| Mean F1 ($\uparrow$) | 67.4 | 68.5 | **68.7** | 67.5 |
| Mean KL ($\downarrow$) | 5.20 | 3.03 | 3.56 | **2.87** |

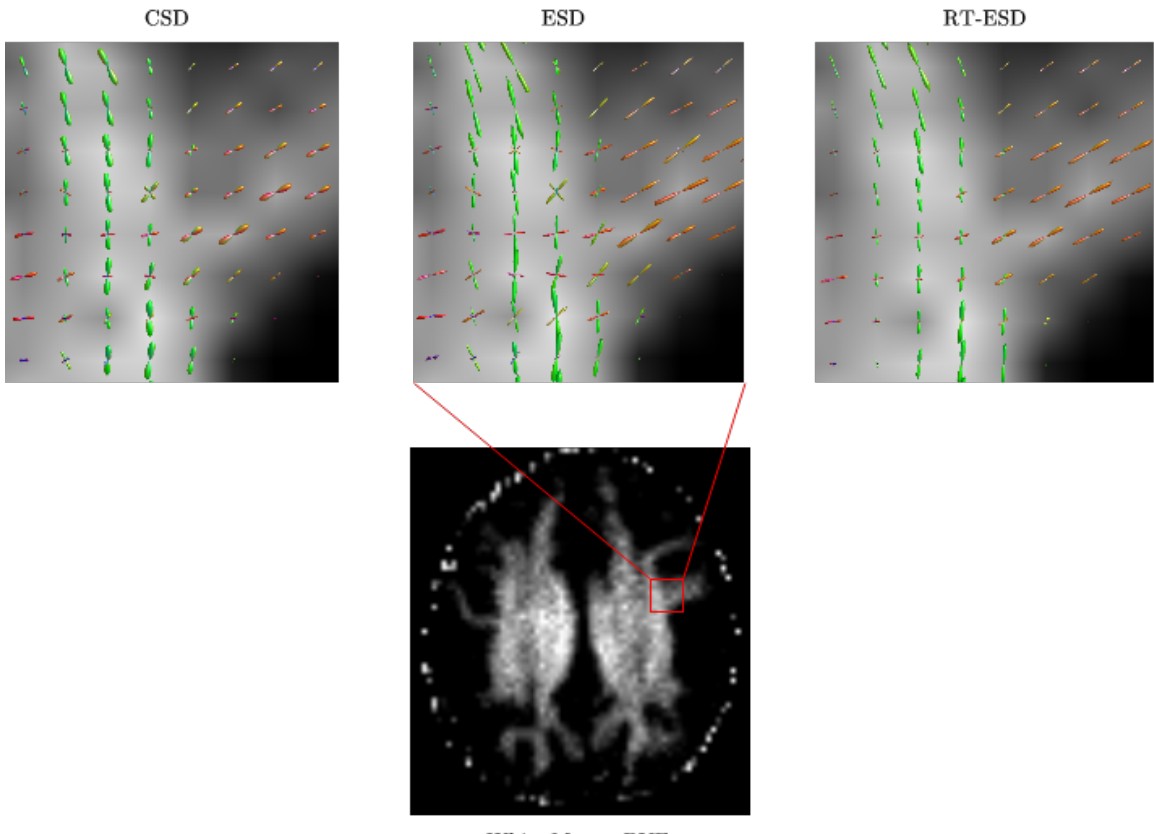

Figure 6: **Estimated fODFs from the Tractometer dMRI dataset.** This figure visualizes results from CSD, ESD, and RT-ESD at a particular location with crossing fibers. RT-ESD yields more spatially-coherent fiber directions with accurate modeling of crossing fibers as compared to the spatially-agnostic ESD and CSD baselines.

**Dataset.** Fiber tracking is a common dMRI task downstream to fODF recovery. However, as in-vivo human dMRI has no associated 'ground-truth' fiber-tractograms, we assess the tractography reconstruction performance from the fODF estimation on the ISMRM Tractometer dataset (Maier-Hein et al., 2017). Tractometer provides a $92 \times 110 \times 92$ simulation of a real human brain, with known fiber configurations. We use the updated version of the scoring data and algorithm whose details can be found in (Renauld et al., 2023). A single shell protocol is used, with 1 B0 image and 32 gradients with a $b$-value of $1000 s/mm^2$. As pre-processing, we only apply dMRI motion-correction using FSL (Andersson and Sotiropoulos, 2016) and divide by the voxel-wise B0 mean. The white matter and grey matter response functions are computed using MRtrix3 (Tournier et al., 2019; Dhollander et al., 2019). We use a two-tissue decomposition fODF estimation. After fODF estimation with all baselines, we generate 100000 streamlines with a minimum length of $60mm$ using MRTrix3 probabilistic tractography with the iFOD2 algorithm using its suggested default parameters.

**Modeling modification.** As opposed to Sections 4.2 and 4.3, we find that Tractometer benefits from a slight modification in its reconstruction objective. Instead of computing the reconstruction loss only on the center voxel of the patch, we find improved results on Tractometer by computing it over the entire spatial patch during training.

**Evaluation Scores.** The Tractometer dataset scoring algorithm compares 21 ground truth fiber bundles against predicted tractograms. We follow Renauld et al. (2023) and compare all methods by computing the number of valid bundles (the number of detected ground-truth fiber bundles), valid streamlines (the proportion of streamlines that are part of a valid bundle), bundle overlap, and overreach (the overlap/overreach between valid predicted bundles and ground truth bundles), and lastly, the F1 scores. We also compute the KL-divergence between the partial volume estimates derived from the predicted fODFs and a probabilistic segmentation of the co-registered T1w MRI provided for the challenge, similar to the human experiments in Section 4.3.

**Results.** As seen in table 3, RT-ESD better estimates the tissue composition of each voxel via the partial volume estimates derived from deconvolution as compared to the other baselines. Simultaneously, it improves the tractography reconstruction in terms of the fraction of valid bundles, valid streamlines, and mean overlap, resulting in an overall more accurate tractography. Qualitatively, in figure 6, RT-ESD yields more spatially-coherent fODFs in comparison to previous works including ESD, CSD, and C-ESD, while also correctly identifying crossing fibers.

## Appendix B. Additional experimental details

### B.1. Implementation details

Adam (Kingma and Ba, 2014) was used for optimization in all experiments. All models were trained on a single NVIDIA A100 GPU. Besides the convolution used, the U-Net architecture (Ronneberger et al., 2015) is the same for the three experiments. It has four average pooling/unpooling and concatenating skip-connections between the encoder and decoder. Two convolutions are applied before each pooling/unpooling. Each convolution is followed by a BatchNorm layer (Ioffe and Szegedy, 2015) and a ReLU activation except for the output layer. The number of filters per layer is doubled after each pooling layer, starting with 16 filters for the $\mathbb{R}^3 \times \mathcal{S}^2$ MNIST experiments and 8 for the dMRI experiments. The spatial kernels have size $3 \times 3 \times 3$. The spherical convolution kernels use a polynomial degree of $K = 5$. For practical stability reasons, we use Chebyshev polynomials $(T^k(\mathbf{L}))_k$ instead of monomials $(\mathbf{L}^k)_k$ (Defferrard et al., 2016). Moreover, as in Perraudin et al. (2019), we use the HEALPix spherical grid (Gorski et al., 2005) with spherical vertices $\mathcal{V} = (p_i)_i$ due to its convenient hierarchical structure. Edge weights are set to $w_{ij} = exp(-\frac{||p_i - p_j||_2^2}{\rho^2})$ if the spherical vertices $i$ and $j$ are neighbors and zero otherwise. Epochs are defined in our context as each voxel in the image having been trained on at least once.

### B.1.1. $\mathbb{R}^3 \times \mathcal{S}^2$ MNIST experiments

To predict the probability of each digit, a Softmax activation is applied to the U-Net output. The inputs are entire $16 \times 16 \times 16 \times 192$ $\mathbb{R}^3 \times \mathcal{S}^2$ MNIST volumes, where 192 is the number

of discrete points on the HEALPix spherical sampling of resolution 4. The model output is a $16 \times 16 \times 16 \times 10$ volume, where 10 is the number of predicted classes. We train each U-Net model for 50 epochs with a batch size of 16. We start with a learning rate of $5 \times 10^{-3}$ and halve it at epochs 25, 35, and 45. We use a joint Dice and Cross-Entropy loss,

$$Loss = 0.5\Big(1 - 2\frac{\sum_N \sum_C w_c(y^n_{c,pred}y^n_{c,true})}{\sum_N \sum_C w_c(y^n_{c,pred} + y^n_{c,true})}\Big) + 0.5\Big(-\sum_N \sum_C w_c\log\frac{e^{y^n_{c,pred}}}{\sum_C e^{y^n_{i,pred}}}y^n_{c,true}\Big),$$

where $w_c = (\frac{1}{\sum_N y^n_{c,true}})^2$ is a class-dependent weight to counter imbalanced labels, $y^n_{c,true}$ and $y^n_{c,pred}$ are the ground-truth label and predicted probability of voxel $n$ and class $c$, $N$ is the total number of processed voxels, and $C$ is the total number of classes.

With respect to network capacity, all models have a roughly equivalent number of trainable parameters. The $E(3)$-equivariant UNets acting on raw channelwise concatenated spherical volumes and spherical harmonics coefficients have 50618 and 41466 learnable parameters, respectively. The voxelwise $SO(3)$-equivariant U-Net has 28682 trainable parameters. Lastly, the proposed $E(3) \times SO(3)$ U-Net has 66090 trainable parameters.

### B.1.2. DISCO AND HUMAN EXPERIMENTS

A final convolution layer and a Softplus activation are applied to the output of the U-Net. The network input is a $P \times P \times P \times B \times 768$ dMRI volume and its output is a $P \times P \times P \times T \times 190$ fODF volume, where $P$ depends on the input patch size (1, 3, 5, or 7), $B$ is the number of shells (4 for both datasets), 768 is the number of spherical pixels using the HEALPix spherical sampling of resolution 8, $T$ is the number of tissue compartments (3 for both datasets), and 190 is the number of even spherical harmonic coefficients used here up to the 18th degree. A spherical harmonic convolution with up to 18 even spherical harmonic coefficient degrees is applied on the fODFs to reconstruct the input dMRI signal. The per-tissue response functions are estimated beforehand with MRTrix3 (Tournier et al., 2019). The convolved fODF and response function results are summed to give a reconstruction of the dMRI input.

We optimize the loss in Section 3. The model is trained for a maximum of 50 epochs with a batch size of 16. The training is stopped earlier if the loss has not improved for 5 epochs. The learning rate is initialized at $1.7 \times 10^{-2}$ and is divided by 10 if the training loss has not improved for 3 epochs. The regularizers used here are taken from Elaldi et al. (2021) and have their weights $\lambda_{\text{sparsity}}$ and $\lambda_{\text{non-negativity}}$ set to $10^{-6}$ and 1.

### B.2. $\mathbb{R}^3 \times \mathcal{S}^2$ MNIST data generation

This section complements Section 4.1 and adds details on how the data was simulated. We construct $16 \times 16 \times 16$ 3D volumes of spherical signals with spatially correlated classification labels in two stages.

First, we construct random classification label volumes for our synthetic volumes. To introduce the spatial dependency between voxels, we first randomly position eight non-overlapping $4 \times 4$ squares on a 2D $16 \times 16$ slice. We then randomly assign a digit between 1 and 9 to the entire square and set the background to the digit 0. We then duplicate this 2D slice along the $z$-axis to get our final 3D classification volume. Note that in the final

volume, we get eight non-overlapping $4 \times 4 \times 16$ *tubes* oriented along the $z$-axis. All voxels have the same classification digit within each tube.

Second, for each voxel, we randomly sample a MNIST digit image corresponding to the classification digit assigned to its square and project it on a sphere as in (Cohen et al., 2018). We use a HEALPix spherical sampling of resolution 4 corresponding to a spherical resolution of 192. As voxelwise spherical digit classification is straightforward, we reduce the information present on one voxel. Instead of projecting the full MNIST image to the sphere, we first randomly crop it to one quarter of its size, and project the cropped digit to the sphere. By doing so, any high-performing learning framework must learn spatio-spherical dependencies. We lastly note that in the grid rotations of this dataset in Section 4.1, the tubes are no longer aligned with the z-axis but have a random direction.

### B.3. DiSCo data preprocessing

This section complements Section 4.2 of the main text. We extract ground truth peak directions using the DiPy peak detection algorithm (Garyfallidis et al., 2014) with a relative peak threshold of 0.5, a minimum separation angle of $25°$, and a maximum number of crossing fibers per voxel of 5. The only pre-processing is a division by the voxel-wise B0 mean. We compute the white matter, gray matter, and CSF response function using MRTrix (Tournier et al., 2019; Dhollander et al., 2019).

