# OpenReview forum: "E(3) x SO(3) - Equivariant Networks for Spherical Deconvolution in Diffusion MRI"
_MIDL.io/2023/Conference — MIDL 2023 Oral_

### Official Review · Reviewer_zuMR · 2023-02-02

**Confidence:** 4
**Preliminary Rating:** 5
**Recommendation:** Best Paper Award

**Summary:**

Diffusion MRI (dMRI) signal in a single voxel can be interpreted as a function on a sphere where each coordinates represent the acquired signal for the given gradient direction. The dMRI signal in a single voxel can be presumed to be correlated with neighbouring voxels as structures may span more than one voxel and multiple structures will appear in the same voxel. Current spherical deconvolution methods for computing fiber ODFs (fODFs), especially neural-network based methods, typically do not take neighbouring voxels into account.

This work presents a novel framework for spherical deconvolution which takes into account the spatial relationship between voxels as well as the spherical nature of the signal, in the form of spherical convolutional network layers. These layers are implemented in a U-Net style architecture trained to reconstruct fODFs from raw dMRI volumes.

Experiments on a heavily-modified version of MNIST, a synthetic phantom and in-vivo subjects are conducted. Results seem to indicate that the method improves on previous methods.

**Strengths:**

The paper shines in its rigour and results. The method is well motivated, experiments are well thought-of and results are convincing. The MNIST experiment is well designed to highlight the strengths of the method. The experiment on the DiSCo dataset is convincing and allows for validation of the method when actual fiber configurations are known. The experiment on in-vivo subjects  that the method works on "real" dMRI.

**Weaknesses:**

While the method is well motivated, it may feel a bit obtuse for readers not familiar with spherical CNNs. Figures 1 and 2 could especially be improved to better display the spatial aspect of the constitutional filters, the actual operations done by the network, the network itself, etc. The text and figures could be improved to better explain how the weight sharing related to neighbouring voxels, the patch size fits into the new layers, for example,

**Deanonymize Review:**

no

**Paper Type:**

methodological development

**Questions To Address In The Rebuttal:**

Please address issues raised in the "Weaknesses" section. The paper could use more intuition and graphical examples. The figures may be improved to put more emphasis on the spatial aspect of the convolutions, and the text may be improved by including a more practical view of the inner working of the layers.

---

### Official Review · Reviewer_uii4 · 2023-02-04

**Confidence:** 3
**Preliminary Rating:** 4

**Summary:**

This paper presents a new technique for the improvement and analysis of diffusion MRI data, called Roto-Translation Equivariant Spherical Deconvolution (RT-ESD). Its main innovation is the inclusion of information from neighboring voxels instead of voxel by voxel analysis. This is evaluated with large public datasets.

**Strengths:**

This is a very neatly written paper with excellent graphicals and text-based explanation of the method.
It features an extensive evaluation section using 2 sets of data (highly sampled multi-shell data and publicly available DiSCo dataset) and a number of evaluation criteria.

**Weaknesses:**

The authors could do an even better job of putting their results in perspective. The discussion section is too short and so is the state of the art section in the background. This is a very active field of research and a number of recent publications exist and should be adequately discussed with their respective limitations and strengths.

**Deanonymize Review:**

no

**Paper Type:**

methodological development

**Questions To Address In The Rebuttal:**

I would love to ask the authors to elaborate on how exactly their method outperforms existing methods. I would alos love to see a more extensive abstract featuring some key information such as improvements observed, datasets used etc.

---

### Official Review · Reviewer_crje · 2023-02-05

**Confidence:** 2
**Preliminary Rating:** 4
**Recommendation:** Poster

**Summary:**

The submission introduces a new deep learning network which is equivariant wrt symmetries of spatial rotations, reflections and translations as well as voxelwise spherical rotations .The new method, RT-ESD, is tested quantitatively on synthetic and challenge data and also qualitatively on human brain BRI. Better accuracy and spatial consistently is achieved compared to the other tested methods.

**Strengths:**

* The submission is well written.
* The submission covers the related work / literature in a detailed way.
* Novelty is related to introduction of E(3)xSO(3) equivariant convolutional networks
* The experiments were carried out on both synthetic and real-world human brain images (where ground truth does not exist): high generalization to unseen poses in the synthetic data, good tissue -specific deconvolution quality


**Weaknesses:**

* The submission seems to heavily build on the framework introduced by Elaldi2021.
* There is a lot of tests / experiments left to establish the method. The experiments so far are promising but very low in number and limited
* in vivo experiments, where ground truth does not exist, are carried out via 3-tissue partial volume estimation

**Deanonymize Review:**

no

**Detailed Comments:**

What is q in Section 2 ?
What is the white arrow pointing at in Figure 4?


**Paper Type:**

both

**Questions To Address In The Rebuttal:**

There are several challenge data sets and tractography data sets that exist with their manual segmentation equivalents. Could the authors discuss how the improved performance on the 3-tissue-segmentation experiments would manifest in the tractography recovery task?

---

### Meta-Review · Area_Chair_VR3i · 2023-02-24

**Recommendation:** Accept (Oral)
**Confidence:** 5

**Metareview:**

The paper presents a new technique called Roto-Translation Equivariant Spherical Deconvolution (RT-ESD) for improving diffusion MRI data. It includes information from neighboring voxels and uses a U-Net style architecture trained to reconstruct fiber ODFs. The method takes into account the spherical nature of the signal and spatial relationships between voxels. The results from experiments indicate that RT-ESD improves upon previous methods in terms of accuracy and spatial consistency.

The submission is well-written and covers related literature in detail. The experiments were conducted on synthetic and real-world human brain images without ground truth, showing high generalization to unseen poses in synthetic data and good tissue-specific deconvolution quality.

The reviewers asked to resituate the work and enrich the discussion in relation to the state of the art, which the authors did and added in the manuscript. Similarly, details were requested about the experiments, especially in terms of ground truth, which the authors did bring during the rebuttal.  Finally, the reviewers asked to add some technical elements on the figures and on the spherical CNN, which the authors have well taken into account and which increases the understanding of their work.

For all these reasons, I recommend that this work be accepted at MIDL.